# Generation of 3D finite element mesh of layered geological bodies in intersecting fault zones

**YingXian Chen, HongXia Yang◉\*, YongChao Ye, JiaYing Li**

College of Mining, Liaoning Technical University, China, Fuxin

\* 3035526032@qq.com

## Abstract

As the geological fault surface divides the 3D space of stratified ores and rocks into complex spatial surface domains, it is necessary to fully consider the spatial relationship between intersecting fault zones and geological bodies in the process of 3D modeling, and how to accurately establish the 3D finite element mesh of geological bodies in intersecting fault zones is a difficult point in modeling complex geological structure. The laminated geological body in intersecting fault zone is a multifaceted domain grid model consisting of a ground-level grid, a geological fault plane grid, and a range grid. By analyzing the spatial relationship between the geological interfaces of the intersecting fault zones, a closed manifold processing method is proposed to establish the closed manifold spatial surface model of the intersecting fault zones, based on which the closed spatial surface model is tetrahedrally divided to establish a 3D solid model. Finally, the 3D solid model is imported into Ansys to generate a 3D finite element mesh. VC++ is used as the development platform for programming, to realize the generation and closed manifold processing of ground level and geological fault surfaces, and use TetGen library to generate finite element mesh based on irregular tetrahedron. Taking an intersecting fault zone in an open-pit mine as an example, the 3D finite element mesh of laminated geological bodies in the intersecting fault zone is established successfully. This method provides an effective and feasible solution for generating accurate 3D finite element meshes in complex stratigraphic spaces based on closed manifold processing.

**Data Availability Statement:** All relevant data are within the paper and its Supporting information files.

**Funding:** The fund includes National Natural Science Foundation of China (52374123) and

## 1 Introduction

Geological faults result in the formation of discontinuous geological bodies and complex fault zones. To accurately analyze the potential geological hazards associated with fault zones, it is crucial to establish a precise 3D geological model of these zones. In open-pit mines containing faults, such as down-dipping layered slopes or reverse-dipping soft rock slopes, the occurrence of landslides is frequent, the probability is high, and effective prevention and control measures are challenging [1–3]. Faults pose a significant hazard due to the pronounced stress concentration that typically occurs in their vicinity. Over time, the presence of a mark can cause alterations in the strength characteristics of the fault itself. Consequently, this gradual

Liaoning Technical University Discipline Innovation Team Project (LNTU20TD-07).

**Competing interests:** NO authors have competing interests.

transformation can ultimately result in diminished slope stability following fault exposure [4, 5]. Landslide accidents in open pit mines associated with faults have been well-documented, underscoring the heightened probability of such incidents near fault zones. This likelihood is further exacerbated in the presence of seismic activity, where earthquakes can significantly increase the susceptibility to landslides. Moreover, it is crucial to consider the size and characteristics of the sliding surface as they exert a profound influence on the safety factor [6]. The dimensions and properties of the sliding surface have a direct impact on slope stability, thereby influencing the overall safety factor associated with potential landslide events [7, 8]. In light of these considerations, it is crucial to generate a more refined grid at the fault to facilitate an accurate analysis of potential hazards associated with the fault in open pit mining. Establishing an accurate 3D geological model is paramount for comprehensively understanding the geological complexities and assessing the risks posed by the fault. By accurately analyzing the hazards induced by the fault, the potential occurrence of landslide geological disasters can be effectively mitigated. This approach ensures a logical and academically rigorous methodology for assessing and managing the risks posed by faults in open pit mines. Most of the current modeling studies have been conducted based on contour lines [9–12]. Existing studies in the field of open pit ore body modeling have primarily focused on modeling individual ore bodies or single faults. However, there has been relatively limited research conducted on the establishment of 3D finite element mesh models specifically for intersecting faults. This represents a gap in the current body of knowledge, as accurately representing the complex spatial relationships and interactions between intersecting fault zones and geological bodies is crucial for a comprehensive understanding of the geological structure. Further research and development of 3D finite element mesh models that account for intersecting faults are needed to address this gap and enhance the accuracy and reliability of modeling complex geological structures in open-pit mining scenarios. Grid models of intersecting fault multifaceted domains generally model each sub-ore body first, which often leads to connection problems between multiple sub-ore bodies [13, 14]. In 3D geological modeling, the establishment of structural models forms the basis for developing attribute models. The process of creating simple layered geological models has reached a relatively mature stage [15]. The main effect of geological faults is that they disrupt the continuity of geological bodies, change the original distribution pattern of stratigraphic data, and form complex geological interfaces with discontinuity and abrupt change characteristics [16, 17]. During the process of 3D geological modeling, the presence of fault zones introduces complexities in the intersection of multiple area models near these faults. The intersecting nature of fault zones adds a layer of intricacy to the modeling process, requiring careful consideration and precise handling to accurately represent the spatial relationships and interactions between different geological domains affected by faulting. The complexity of extracting stratigraphic modeling data in areas containing intersecting faults is exacerbated by factors such as the proximity of fault data, sparse data availability, and the challenges involved in data acquisition [18, 19]. Consequently, these factors contribute to increased difficulty in generating tetrahedral mesh in such complex areas. The accurate representation of intersecting faults within the mesh becomes challenging due to limited and fragmented data, requiring specialized techniques and methodologies to overcome these obstacles to achieve a reliable and precise mesh generation process [20, 21]. The fault model describes the contact relationship and geometry between faults, and the construction of the fault model is based on the establishment of a 3D geological model to provide constraints for the generation of the ground level [22, 23]. The construction of complex fault surfaces and the establishment of topological relationships of the elements of the geological body, that is, the establishment of topological relationships between intersecting fault surfaces, between fault surfaces and ground levels, and stratigraphic surfaces, are the core of the entire 3D modeling [24, 25]. Simultaneously, the

spatial relationship of the geological interface within the intersecting fault zone, along with the closed manifold process, serves as the fundamental basis for modeling. It is upon this foundation that the generation of the 3D finite element mesh can be achieved [26, 27]. For improved finite element analysis of geological bodies, precise mesh generation of the modeling area is necessary when constructing the geological solid model. In the modeling process, a complex geological space is comprised of numerous interconnected geological grid surfaces [28–30]. The correct topological connectivity relationship between each grid surface is typically a prerequisite for modeling, model editing, Boolean operations, or even 3D model analysis [31]. A. L. TERTOIS [32] et al employed discrete smoothing interpolation to develop a tool that enables small, real-time adjustments to faults in tetrahedral models. Qiang Tianchi [33] implemented a transition between 3D tetrahedral sparse and dense meshes by coupling two-part cells with an interface transition cell, using the principle of minimum potential energy to ensure coordination of the displacements and strains of each corresponding point on either side of the interface. Deyun Zhong [34] et al. introduced an adaptive mesh partitioning technique that separates the interstitial domain using feature constraints on the contour lines of the ore body or fits the intermediate domain using a distance field. Zhou [35] et al. studied the unstructured mesh generation method in finite element preprocessing and its application in mining, proposing the adaptive mesh segmentation method, which segregates or merges intermediate domains through the distance field and applies feature constraints to the contours of the ore bodies. Hang Si [36, 37] et al. used the classical Delaunay refinement algorithm to build a TetGen library to generate an isotropic tetrahedral mesh. Souche L [38] et al. based on a global interpolation method for data extraction from faults and strata, capable of constraining any surface by all valid conformal data without being constrained by its type. Godefroy [39] used numerical fault operators to make the strata fall according to a theoretically isolated fault displacement model and produce consistent fault displacements, to make the structural model consistent. Lobatskaya [40] et al. used a finite element model to reconstruct the stresses within the block around the inclined fracture in the simulated fault damage zone. Feng [41] et al. determined the safety factor of the slope by creating a 3D numerical model and simulating the slope model with the finite element strength discount method, which allowed the identification of the specific damage mode of the slope during production. Sun Shiguo [42] et al. established a three-dimensional model that calculated the displacement and stress of the slope body by the finite difference method and investigated the characteristics of the influence of multi-stage open-pit cascade mining on the stability of the slope and its slope displacement and stress distribution. Chen [43] et al. created a three-dimensional solid model of the rock mass that was blasted by utilizing the lithology data as a sample. They then employed this model to compute the quantity of charge present in the blasted rock mass. Modeling individual faults was comparatively simple and straightforward. However, the modeling process becomes inherently complex when faults exhibit different tendencies, such as interlacing, overlapping, and truncation between fault surfaces.

In summary, researchers have made a lot of achievements in the research of 3D geological modeling with faults, but there are fewer studies on 3D finite element mesh generation of open pit slopes with intersecting fault zones, and the models built have problems such as insufficient accuracy and low efficiency. To better solve the existing problems, from the consideration of accurate geological 3D solid modeling of complex open pit slopes containing faults, this paper tries to do some exploratory work based on the previous research and propose a 3D finite element mesh generation method for laminated geological bodies containing interlaced fault zones based on the manifold processing of interlaced mesh surfaces, which provides a new method for finite element mesh generation of complex geological models.

## 2 Spatial relationship of geological interfaces of intersecting fault zones and treatment of closed manifold

In the construction process of 3D finite element meshes for complex stratigraphic spaces, the presence of interlaced constraint surfaces among faults, faults, and various strata, as well as among different strata, creates a non-connected relationship, posing challenges in generating a cohesive 3D finite element mesh for the entire system. The modeling complexity and difficulty are significantly increased due to the intricate intersection of complex grid structures. To generate 3D finite element grids with complex stratum constraint surfaces, it is necessary to study the spatial relationship of non-manifold strata constrained by multiple areas, eliminate the influence of non-connectivity and non-manifold of 3D spatial grids, and generate connected closed grids.

### 2.1 Closed manifold grid for complex 3D geological space

Complex 3D geological space consists of multiple independently generated geological grid surfaces, with each surface being in a non-connected state. To establish connected closed grids, it is essential to adhere to the rules of connected grids, manifold grids, and closed grids. A connected mesh refers to a polygon mesh where any two vertices are linked by a path consisting of an edge, as shown in Fig 1. A polygonal mesh is a collection of vertices, edges, and faces which must satisfy the following three points: ① Each vertex must share at least one edge. ② Each edge must share a face. ① If the polygon mesh faces intersect, they must intersect their vertex or face (the two faces cannot intersect each other).

Manifold mesh refers to a sufficiently small neighborhood that exists at any point on the surface of a 3D solid, which is isomorphic to a disk on that surface. In other words, there is a continuous one-to-one mapping between this neighborhood and the disk, enabling the shape to form a complete and non-overlapping plane after any arbitrary distortion transformation [44]. For any given shape, if it is non-empty in 3D Euclidean space R3, has a boundary that is fully enclosed, and its boundary is a two-dimensional manifold, then it is considered a regular shape, otherwise, it is classified as a non-manifold shape or a non-regular shape [44]. A manifold mesh must satisfy the following three conditions: ① It must be a connected mesh; ② Each edge must have only two shared faces; ③ Multiple faces associated with a vertex and that vertex form a closed or open sector, as shown in Fig 2, in which the closed sector is shown in Fig 2(a) and the open sector is shown in Fig 2(b). If the above conditions are not satisfied, it is a non-manifold mesh.

In solid geometric models, a triangular mesh can be represented as a simple complex form composed of three tuples $(v,e,f)$. In a closed and interconnected manifold triangular mesh,

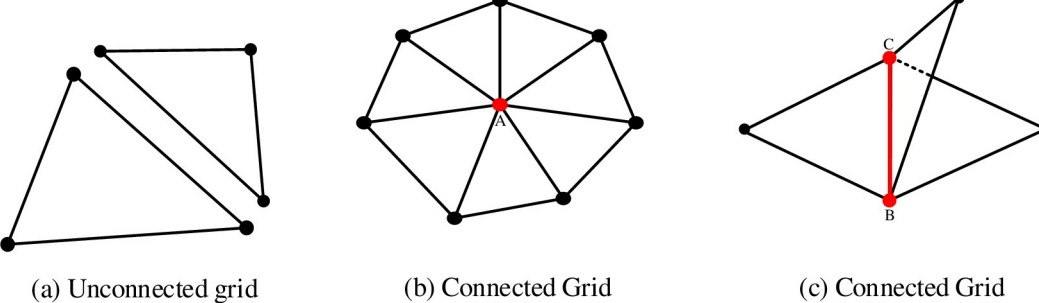

(a) Unconnected grid (b) Connected Grid (c) Connected Grid

**Fig 1. Connected grid and unconnected grid.** The red point A and red line segment BC in (b) and (c) of the figure are connected points and lines.

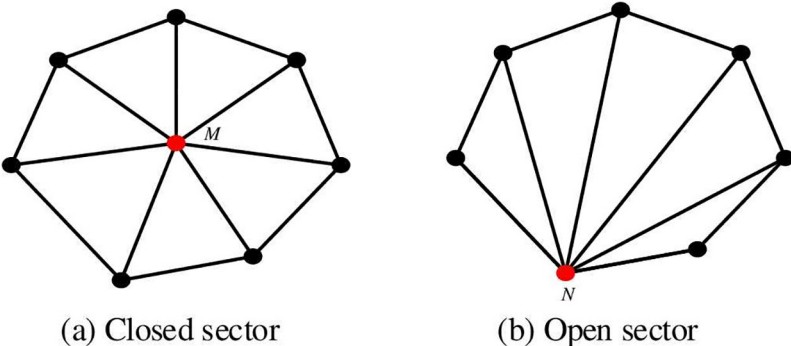

**Fig 2. Connected closed sector and open sector.** Where the red points M and N are the incident vertices.

there exists: an Euler relation that involves the relationship between the number of vertices, edges, and faces $v+e+f = 2$. Wherein $v$, $e$, and $f$ are the number of vertices, edges, and faces, respectively. Set a vertex set $V = \{v_1, v_2 \ldots, v_i, \ldots, v_n\}$, which determines the geometry and position of the mesh in the 3D European space.

$E = \{e_1, e_2 \ldots, e_i, \ldots, e_n\}$, $e_i \subset V * V$ represents the set of edges connected by 2 vertices. $F = \{f_1, f_2 \ldots, f_i, \ldots, f_n\}$, $f_i \subset V * V * V$ represents the set of triangular faces connected by 3 vertices. Based on the above relationships, the following statistics can be derived: (1) The number of triangular surfaces is approximately twice the number of vertices, $f \approx 2v$. (2) The number of edges is approximately equal to three times the number of vertices, $e \approx 3v$ The average number of vertex keys (*numberofincidentedges*) is 6. A closed manifold mesh refers to a manifold mesh where each vertex has a closed fan, the mesh has no boundaries, or each edge of the mesh is shared by only two faces. Such a mesh is called a closed manifold mesh (except in cases where the mesh is self-intersecting). Therefore, for any partitioned closed manifold mesh, the Euler relation holds. In the process of modeling complex 3D solids, there are generally multiple triangles combined to form a layer of mesh surfaces, and complex spatial relationships will be formed between the mesh surfaces. And very complex non-manifold situations will be encountered in the interlaced mesh surfaces in space, such as suspension lines, suspension surfaces, spatial intersections, and so on, as shown in Fig 3.

## 2.2 Spatial relationships of intersecting fault zones and geological interfaces

When intersecting faults exhibit cross-cutting, truncating, or overlapping relationships, various truncation patterns can emerge, such as $Y$—shaped, $\chi$—shaped, $\lambda$—shaped, semi –$Y$– shaped, and semi –$\lambda$– shaped truncations [45]. The analysis of spatial relationships between intersecting fault surfaces, fault surfaces, stratigraphic surfaces, and stratigraphic surfaces and geotechnical interfaces is a prerequisite in the modeling process.

**2.2.1 Spatial relationship between the grid of intersecting fault surfaces.** Different spatial relationships exist between the grids of complex fault surfaces, while the spatial relationships of the grids of intersecting fault surfaces are the most complex. In the chronological order of interlaced fault generation, the newly generated faults will divide the old faults into several parts. Taking a specific open-pit mine in Inner Mongolia Autonomous Region as an example, there are two faults within the geological formation, namely $F_1$ and $F_2$, both of which are normal. The vertical displacement of $F_1$ is 25m, while $F_2$ has a displacement of 30m. Furthermore, $F_2$ is formed after $F_1$, and it divides $F_1$ into two segments known as $F_{1U}$ and $F_{1D}$. The

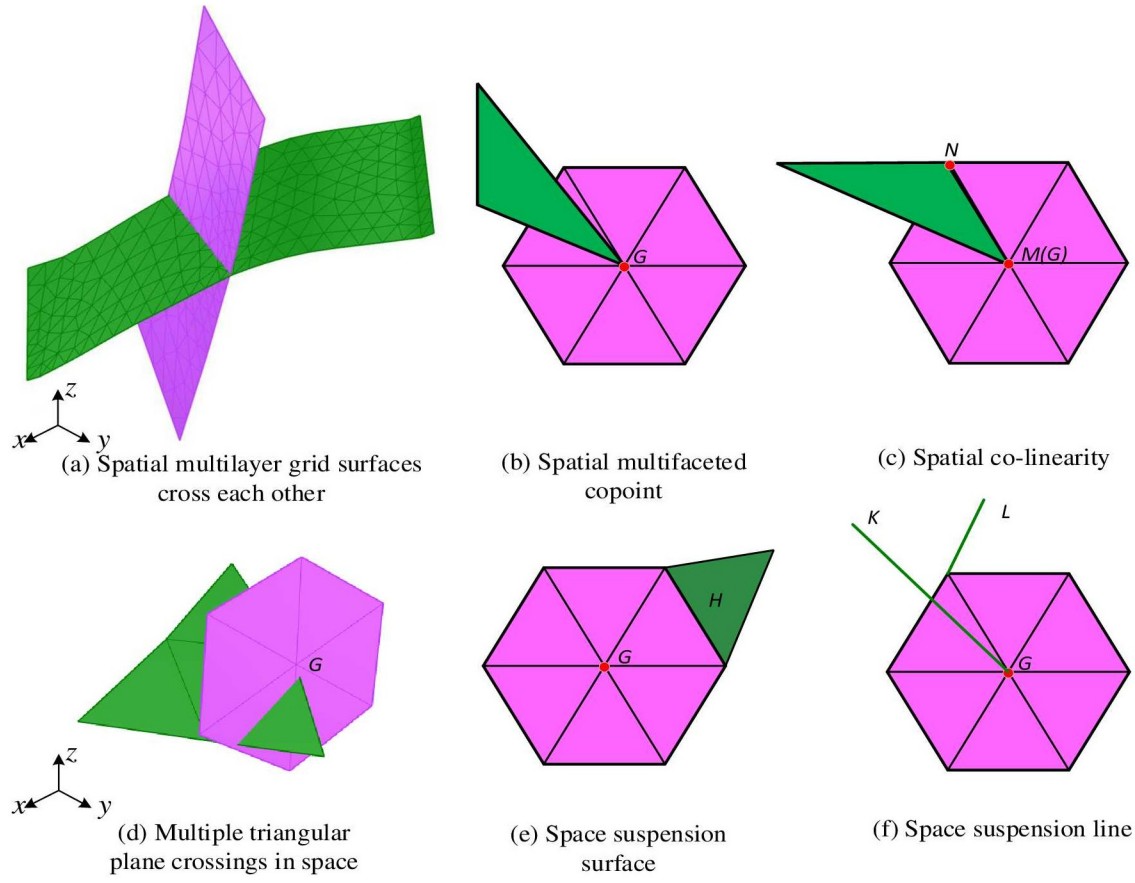

(a) Spatial multilayer grid surfaces cross each other

(b) Spatial multifaceted copoint

(c) Spatial co-linearity

(d) Multiple triangular plane crossings in space

(e) Space suspension surface

(f) Space suspension line

**Fig 3. Several common non-manifold forms.**

fault interface grid is a triangular mesh surface generated by fitting exploration borehole data and fault parameters such as dip, strike, and fault trace, as shown in Fig 4. Among them, the mesh surface of fault $F_2$ exhibits a non-connected spatial relationship with the two mesh surfaces of fault $F_1$, namely $F_{1U}$, and $F_{1D}$.

**2.2.2 Spatial relationship between fault surface mesh and stratum interface mesh.** The strata on either side of a fault undergo relative displacement along the fault surface, resulting in the formation of the upper and lower strata interfaces of the fault. The stratigraphic interface grid is a triangular grid generated by regional interpolation based on exploration drilling data, cross-section profiles, and fault intersections. The positions and spatial relationships between the fault surface grid and the stratum interface grid are shown in Fig 5. In this figure, the intersecting faults $F_1$ and $F_2$ divide the stratum interface into four parts, namely $P_1$, $P_2$, $P_3$, and $P_4$. The spatial relationship between the fault surface grid and the stratum interface grid is as follows:

(1) Part $P_1$ is located in the lower plate of faults $F_1$ and $F_2$, and the stratigraphic interface grid of this part intersects with fault surface grids $F_{1D}$ and $F_2$, respectively, which belong to the non-connected spatial relationship.

(2) Part $P_2$ is located in the lower plate of fault $F_1$ and upper plate of $F_2$, and the stratigraphic interface grid of this part intersects the fault surface grids $F_{1U}$ and $F_2$, respectively, which belong to the non-connected spatial relationship.

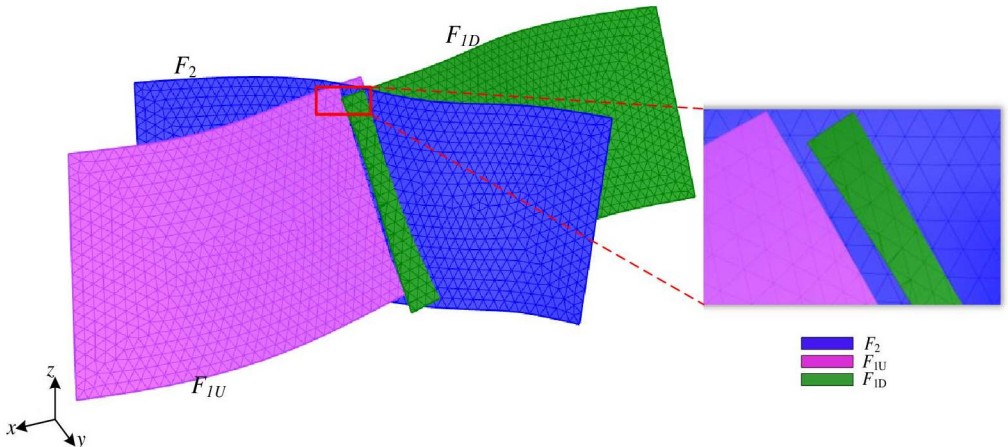

**Fig 4. Spatial relationship of the grid of intersecting fault surfaces.** The red boxed part is the map after the fault slip, and the right figure shows its enlarged part.

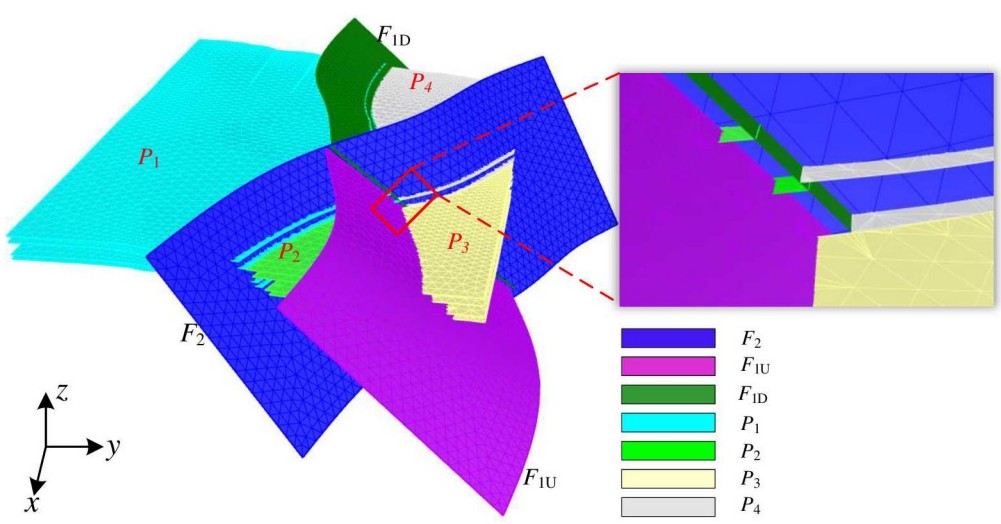

**Fig 5. Grid spatial relationship between faults and stratigraphy.** The right figure shows the enlarged view in the red box.

(3) Part $P_3$ is located on the upper plate of faults $F_1$ and $F_2$, and the stratigraphic interface grid of this part intersects the fault surface grids $F_{1U}$ and $F_2$, respectively, which belong to the non-connected spatial relationship.

(4) Part $P_4$ is located in the upper plate of fault $F_1$ and lower plate of $F_2$, and the stratigraphic interface grid of this part intersects with the fault surface grid $F_{1D}$ and $F_2$ respectively, which belongs to the non-connected spatial relationship.

**2.2.3 Spatial relationship between the range grid and the geological interface of the fault zone.** The range grid is a closed mesh formed by combining the boundary surfaces and the bottom surface with the geological interface of geotechnical engineering. It forms complex

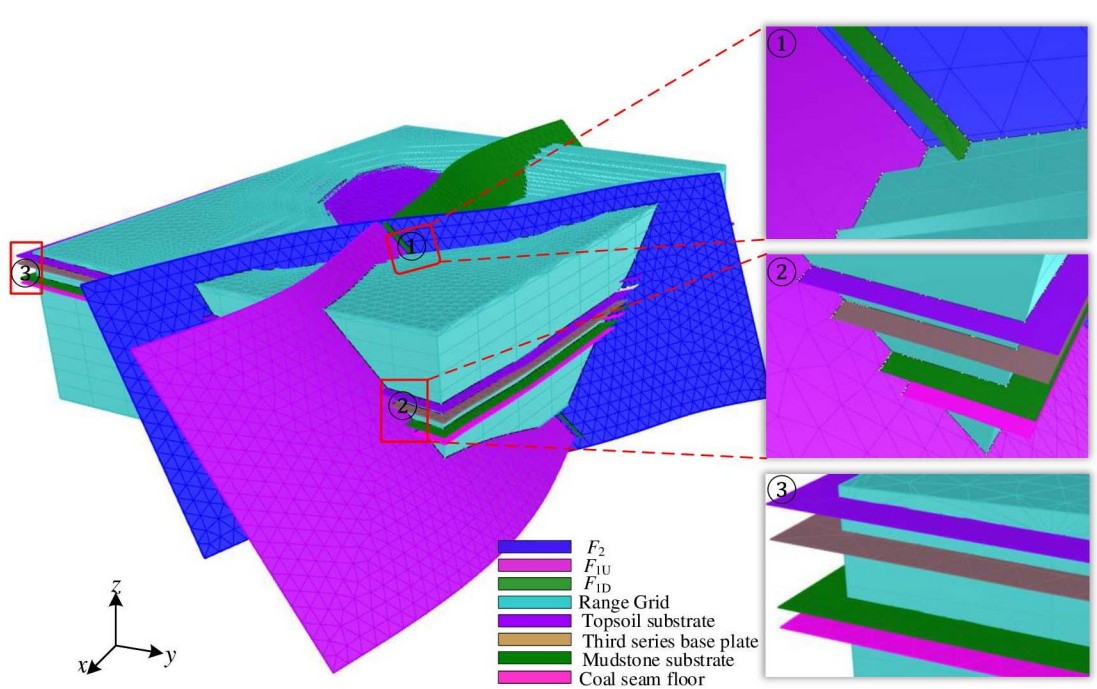

**Fig 6. Spatial relationship between geotechnical interface and geological interface of fault zone.**

spatial relationships with the fault surface grid and the stratum interface grid. The range grid intersects with the fault surface grid and the stratum interface grid, forming a non-connected spatial relationship, as shown in Fig 6.

Among them, Fig 6 (①) shows the intersection of the range grid with the intersecting fault surfaces $F_{1U}$, $F_{1D}$, and $F_2$. Fig 6 (②) illustrates the intersection of the range grid with the intersecting fault surface $F_{1U}$ and the stratum interface grid. Fig 6 (③) demonstrates the intersection of the range grid with the stratum interface.

## 2.3 Spatially closed manifold of geological interfaces in intersecting fault zones

According to the principle of spatial division, any complex shape of an object can be simulated by a finite number of simple shapes. In the spatial domain of an interleaved fault zone, due to the mutual partitioning among fault interface meshes, stratigraphic interface meshes, and boundary meshes, a complex and disconnected space is formed. According to the principle of spatial decomposition, it can be subdivided into multiple simple closed manifold spaces. The key to achieving the closure and manifold representation of geological interface spaces in interleaved fault zones lies in the constraint processing of intersecting mesh faces to ensure consistent topological structures. The following are the methods for constraint processing of intersecting mesh faces.

Step 1, find the intersection lines between the intersecting grids.

Step 2, triangles are generated according to the Delaunay criterion with the intersection lines as constraint edges and added to the intersection mesh, respectively.

In step 3, the grid elements outside the boundary in the intersecting grid are deleted with the intersection line as the boundary respectively to form a new geological interface grid.

The constraint processing of intersecting grid faces is performed sequentially between two intersecting grid surfaces, following an order from inner to outer, top to bottom, or bottom to top. The specific processing steps are as follows: first, perform constraint processing between fault grid surfaces; in the order from top to bottom, apply constraint processing between each layer grid surface and the corresponding fault grid surface; finally, sequentially apply constraint processing between all geological interface grid surfaces and the range grid surface.

**2.3.1 Constraint processing between intersecting fault surface meshes.** The fault surface grids $F_{1U}$ and $F_{1D}$ intersect with $F_2$, respectively, as shown in Fig 7(a). Taking the constraint processing of grids $F_{1D}$ and $F_2$ as an example, the constraint processing process between the fault surface grids is described. Firstly, the grid surface $F_{1D}$ is intersected with $F_2$ to determine the intersection line between the two grids, as shown in Fig 7(b). The intersection line is then used as a constraint edge to triangulate the $F_{1D}$ and $F_2$ meshes individually. This process produces modified $F_{1D}$ and $F_2$ meshes that incorporate the intersection line, ensuring topological consistency between the intersecting meshes, as shown in Fig 7(c). Finally, the grid elements of grid $F_{1D}$ outside the intersection line are deleted to obtain the constrained processed grids $F_{1D}$ and $F_2$, as shown in Fig 7(c). Using the newly generated fault mesh $F_2$, the same method and process are implemented to process the constraints of mesh $F_{1U}$ and $F_2$, and the finished process is shown in Fig 7(d).

**2.3.2 Constraint processing between fault grid and stratigraphic interface grid.** The fault grids $F_{1U}$, $F_{1D}$, and $F_2$ divide the stratigraphic interface into four regions, $P_1$, $P_2$, $P_3$, and $P_4$, respectively, as shown in Fig 8(a). $P_1$ is located between the lower segment of the $F_{1D}$ fault grid and the lower segment of $F_2$. $P_2$ is located between the lower segment of the $F_{1U}$ fault grid and the upper segment of $F_2$. $P_3$ is located between the upper segment of the $F_{1U}$ fault grid and the upper segment of $F_2$. Finally, $P_4$ is located between the upper segment of the $F_{1D}$ fault grid

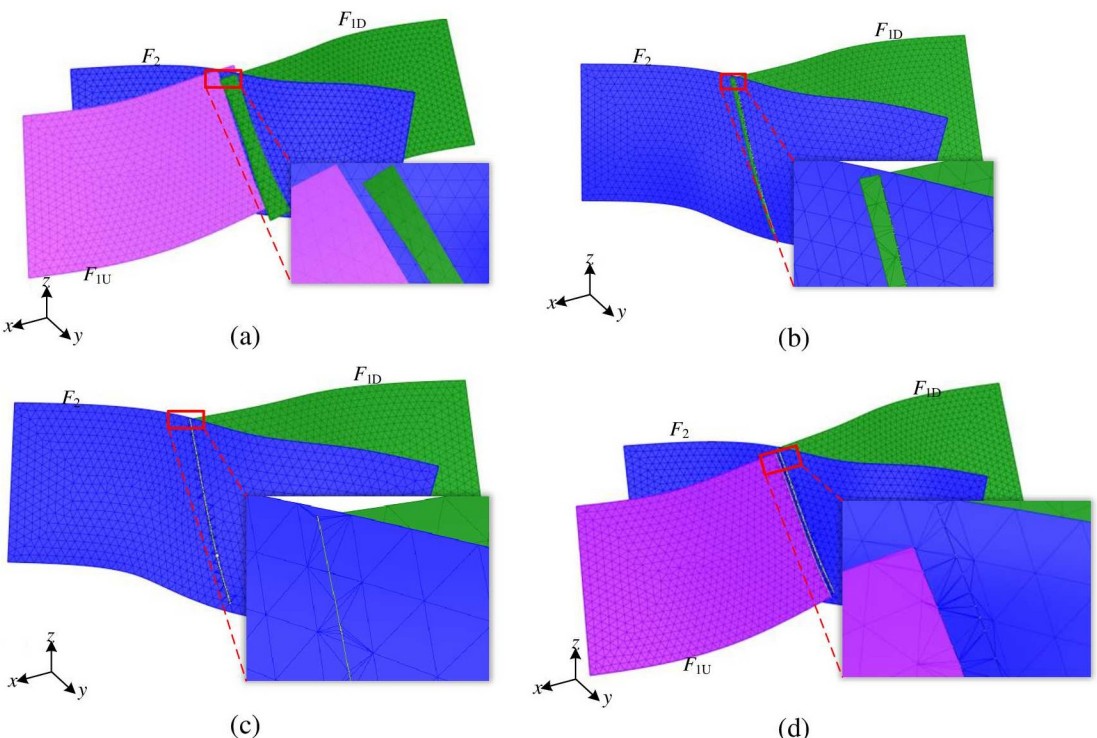

**Fig 7. Spatial relationship between geotechnical interface and geological interface of fault zone.**

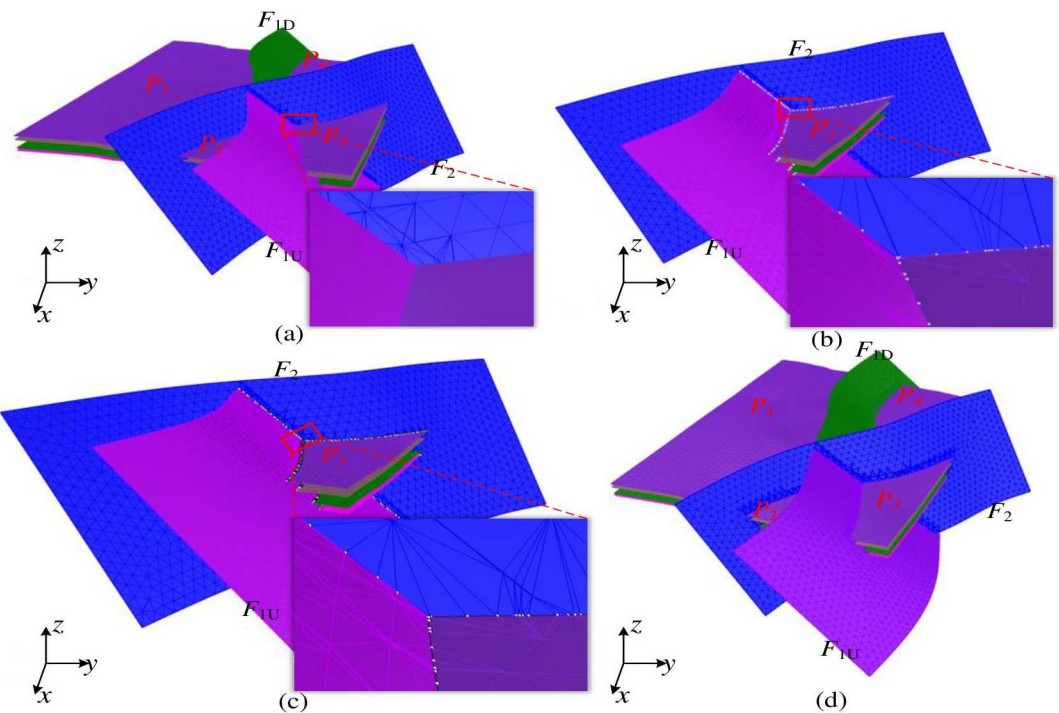

**Fig 8. Constraint processing of intersecting faults and stratigraphic grid.** The bottom right corner shows the enlarged part of the red box. The black line indicates the intersection line of the two faces, and the white dots indicate the required corresponding topology line of the face, we can judge whether the current topology of the two faces is consistent or not based on the white dots.

and the lower segment of $F_2$. To illustrate the process of constraint processing between intersecting fault grids and stratigraphic plane grids, the constraint processing between the $F_{1U}$ and $F_2$ fault grids and the stratigraphic interface grid in the $P_3$ area is shown as an example. First, the $P_3$ stratigraphic interface grid is intersected from top to bottom with the $F_{1U}$ fault grid. After obtaining the intersection line, it is used as a constraint edge to triangulate both the stratigraphic interface grid and the $F_{1U}$ fault grid. The results of the constraint processing are shown in Fig 8(b). Secondly, the same approach is used to perform constraint processing between the stratigraphic interface grid and the fault grid $F_2$ in the $P_3$ region. The results are shown in Fig 8(c) (The black-white dotted line is the line of intersection between faces). Finally, the parts of the stratigraphic interface grid outside the intersection of the $F_{1U}$ and $F_2$ fault grids are removed to obtain the constrained processed grid. The stratigraphic interface grids of other regions are sequentially processed using the same method, and the constrained processed stratigraphic interface grids and fault grids for each region are shown in Fig 8(d).

**2.3.3 Confined constraint processing between the range grid and the geological interface grid of the fault zone.** The range grid T intersects with all the geological interface grids of the fault zone, as shown in Fig 9(a). To achieve a closed manifold grid, it is essential to enforce constraint processing between the range grid and the geological interface grids. The constraint processing between grids follows a sequential order from inner to outer and from top to bottom. This means that the intersection between the range grid and the fault grids is processed first, followed by the intersection between the stratigraphic interface grids of each region and the range grid in a top-down manner. Firstly, the fault grids $F_{1U}$, $F_{1D}$, and $F_2$ are intersected with the range grid, and the intersection lines are used as constraint edges to

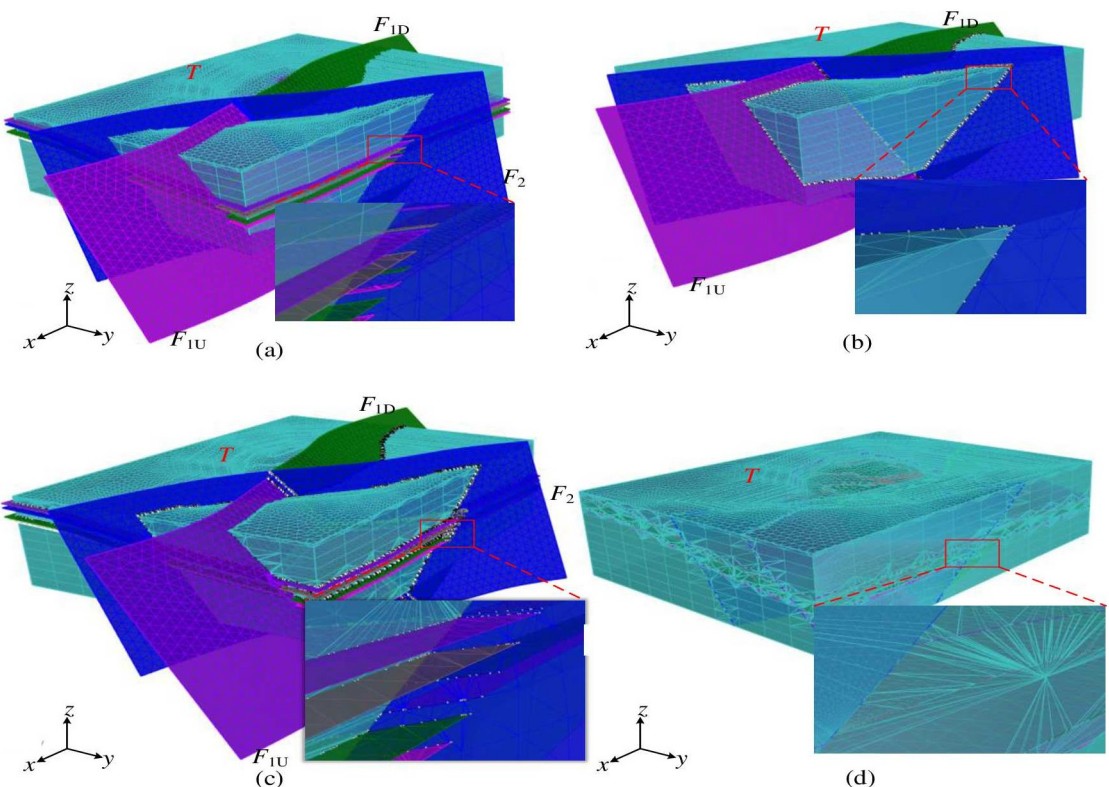

**Fig 9. Confined processing of the geological interface between the range grid and the fault zone.** The bottom right corner shows the enlarged part of the red box. The black line indicates the intersection line of the two faces, and the white dots indicate the required corresponding topology line of the face, we can judge whether the current topology of the two faces is consistent or not based on the white dots.

perform triangular dissection and constraint processing on the fault grids and the range grid, as shown in Fig 9(b). Secondly, the stratigraphic interface grid of each region is intersected with the range grid from top to bottom. The intersection line is used as a constraint edge for each stratigraphic interface grid and the corresponding range grid, allowing constraint triangulation. This process continues until constraint processing is complete for all stratigraphic interface grids and their corresponding range grids. The resulting constrained grids are shown in Fig 9(c). Finally, the fault grid and all the grid elements of the stratigraphic grid located outside the range grid are removed, resulting in the achievement of a spatially closed manifold for the geological interface of the intersecting fault zone. This process generates multiple closed manifold grids, as shown in Fig 9(d).

## 3 3D finite element mesh generation for laminated geological bodies in intersecting fault zones

### 3.1 Establishment of a 3D solid model of the closed manifold space of the geological interface of the intersecting fault zone

The generation of tetrahedral mesh in 3D solid modelling is the prerequisite and difficulty of FEA calculation, the specific modelling steps are as follows: (1) Establishment of a borehole database using engineering geological information in the region; (2) Grids of slope surfaces, geological interfaces, and fault surfaces were created through Delaunay triangular profiles

interpolation using the inverse distance method; (3) Create a range grid and combine it with the slope surface grid to form a new range grid; (4) The new range grid is restricted to the level grid of each location, resulting in a closed, connected, and topologically consistent spatial surface domain; (5) Mesh optimization of the triangular mesh model underwent optimization using the Triangle 1.6 program; (6) Finally, using Visio Studio 2012 development platform, the TetGen library is developed to achieve the intersecting fault open pit mine slope mesh tetrahedral dissection to generate tetrahedral mesh, and the generated tetrahedral mesh is optimised.

**3.1.1 Spatial grid optimization of geological interface closed manifold.** Mesh optimization is divided into geometric shape optimization and topological relationship optimization. Directly utilizing the closed manifold grid of the geological interface in the intersecting fault zone for a tetrahedral generation often results in the creation of long and narrow tetrahedra, wedges, and other tetrahedral elements with suboptimal shape and quality, as shown in Fig 10. The inferior quality of the generated mesh will adversely affect the accuracy of 3D finite element analysis and simulation. Consequently, it becomes essential to optimize meshes with subpar shape quality. First, all the quadrilateral meshes of the closed manifold space of the geological interface are converted into triangular meshes. Then, the mesh is optimized using Jonathan Richard Shewchuk's triangle 1.6 programs (the minimum angle is 30˚ and the maximum angle is 130˚ after optimization), the pseudo-code for the triangle optimization is shown in Fig 10, and the optimized mesh is shown in Fig 11.

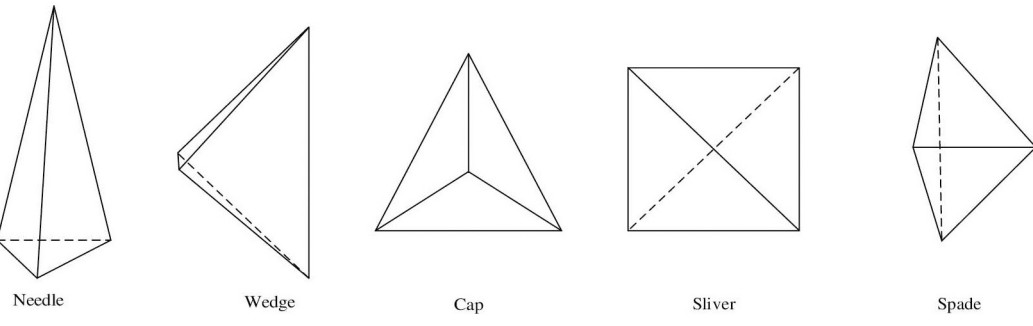

**Fig 10. Several tetrahedral elements with poor shape quality.**

```
Algorithm 1: Triangle optimization
    Input  : Triangular mesh
    Output : Triangular mesh
  1  items ←(add) angle
  2  items ←(add) area
  3  if b.quality(m.triangles.items) > 0 then
  4  |   Enforce quality(m, b)
  5  |   Angle constraint          \\If (b) mass and (m triangle term >0), enforce
  6  |   Area constraint           mass (&m,&b), Angle and area constraints.
  7  End
```

**Fig 11. Triangle optimization pseudo-code.**

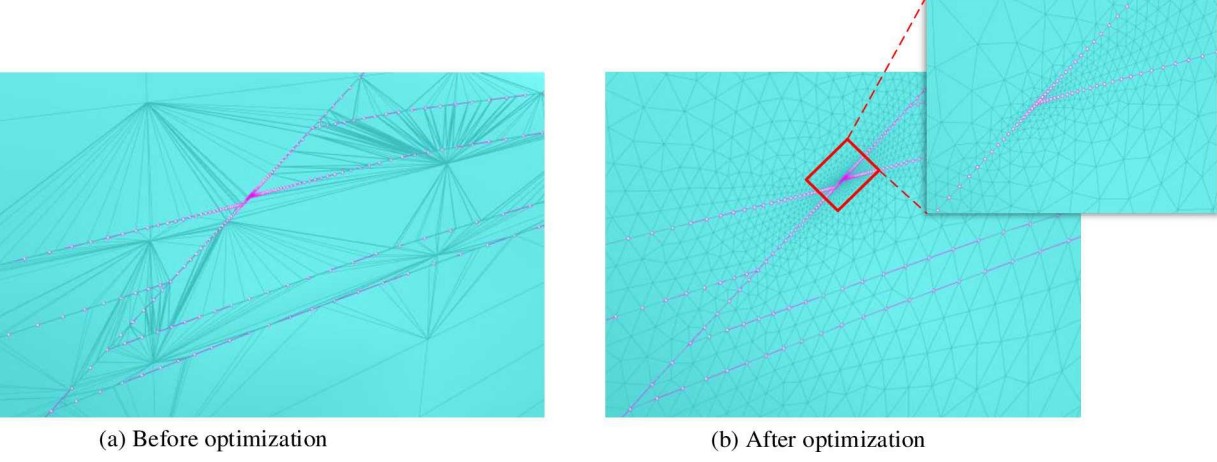

(a) Before optimization  (b) After optimization

**Fig 12. Comparison chart before and after grid optimization (partial).**

The optimized mesh is shown in Fig 12.

Finally, after performing geometric optimization, the mesh is adjusted to address any local topological inconsistencies and ensure complete topological consistency, as shown in Fig 13.

**3.1.2 Irregular tetrahedral solid model based on TetGen library raw component region.** A tetrahedral solid model generation and optimization program was developed using Visual C++2012 based on the TetGen library, and the tetrahedral optimization pseudo code is shown in Fig 14. To resolve the issue of local optima in the TetGen library tetrahedral optimization "local optima", we integrate the procedure depicted in Fig 15 into the optimization process. The closed grid model of the intersecting fault zone was partitioned into tetrahedra to create an irregular tetrahedral solid model representing the intersecting fault zone, as shown in Figs 14 and 15. Fig 16(a) illustrates the closed-connected grid model of the intersecting fault zone, while Fig 16(b) showcases the irregular tetrahedral solid model and its internal structure. Additionally, the tetrahedral solid models of different regions within the intersecting fault zone are distinguished using various colors.

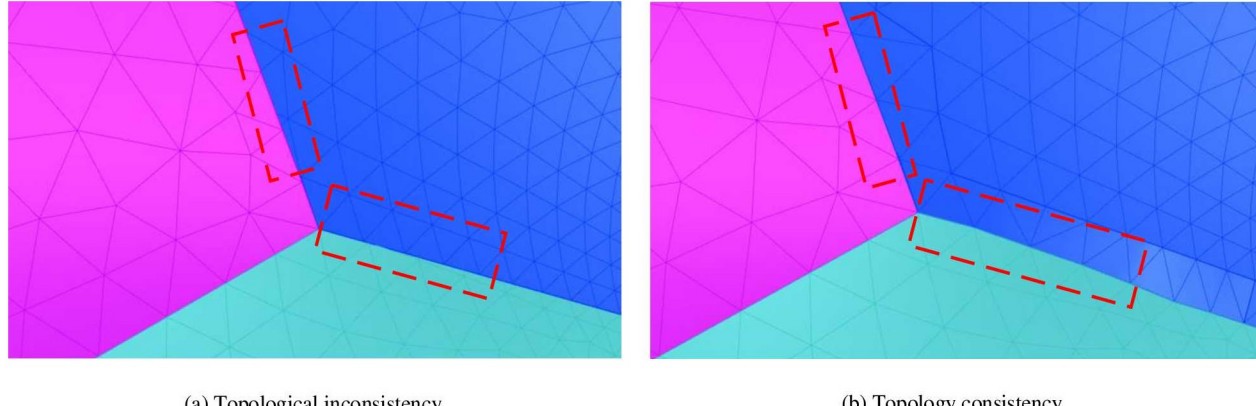

(a) Topological inconsistency  (b) Topology consistency

**Fig 13. Topology adjustment before and after topology adjustment.** The dashed box shows the comparison before and after topology modification.

```
Algorithm 2: Tetrahedral optimization
Input   : T={t_0, t_1, ..., t_i, ..., t_n}. C={c_0, c_1, ..., c_i, ..., c_n}.
          V={v_0, v_1, ..., v_j, ..., v_m}. E={e_0, e_1, ..., e_r, ..., e_k}.
Output: The new tetrahedron
1  for i = 1 to n do
2      b_i = α ρ/h_max                          \\Calculate the mass of the tetrahedrone.
3      for i = 1 to n do
4          if b_i < quality then                \\Optimize when the mass is less than the set
5              delaunay refinemen               mass
6              for i = 1 to n do
7                  flip the face                \\The flip face operation replaces the old
8                  compute b_i'                 tetrahedron when the tetrahedron quality
9                  while b_i' > quality do      improves, and enters the edge flip at the
10                     b_i' replace b_i          end of the flip
11                     break

12              for r = 1 to k do               \\The flip edge operation replaces the
13                  flip the edges              old tetrahedron when the mass of the
14                  compute b_r''               tetrahedron improves, and enters the
15                  while b_r' > quality do      vertex translation operation at the end
16                     b_r'' replace b_i         of the flip
17                     break

18              for j = 1 to m do               \\Vertex translation, replacing the old
19                  vertex translation          tetrahedron when the mass of the
20                  compute b_j'''              tetrahedron increases, and ending the
21                  while b_j' > quality do      loop when the flip ends
22                     b_j'' replace b_i
23                     break

24          else                               \\Otherwise, when the mass is greater than the
25              continue                        set mass, optimize the next tetrahedron

26  End
```

**Fig 14. TetGen library for optimizing tetrahedral pseudo-code.**

## 3.2 Conversion of the 3D solid model to finite element mesh

The three-dimensional geological solid model is combined through the interface of each rock layer to form a closed solid model, and the physical and mechanical parameters of the rock body are assigned to each geological entity, and the built irregular tetrahedral solid model is exported in the .cdb file format. The .cdb file was then imported into Ansys (Ansys Workbench Workbench, 2019) software for meshing, and the lithological parameter properties and contact properties of each layer were set to complete the construction of the 3D finite element model. Finally, the lithologies for each formation are assigned based on lithological parameters stated in Table 1. The lithology sequence, from top to bottom, includes topsoil, tertiary, mudstone, coal, and mudstone. This results in the conversion of the model into a 3D finite element mesh with interactive control, as shown in Fig 17.

## 4 Conclusion

Through the analysis of the intricate spatial relationships among different types of meshes in the intersecting fault zone, a closed manifold processing method is proposed. This method aims to establish a closed manifold spatial mesh model of the intersecting fault zone, followed

---

**Algorithm 3:** Tetrahedral optimization

 **Input** : Unoptimized tetrahedral mesh
 **Output**: Optimized tetrahedral mesh

1 **if** $n = 3$ **then**
2 **if** *remove e* **then** \\If n = 3, when e is removed, it returns the current size
3 return A=m of A to m, otherwise it cannot be flipped to continue.
4 **else**
5 it is not flippable
6 **while** $n > 3$ **do**
7 **for** $i = 0$ *to* $n - 2$ **do** \\If a face in F successfully completes the
8 $F \overset{remove}{\leftarrow} e$ flip, |A| decreases by 1. When a face in F
9 |A|-1 cannot be removed, end.
10 **if** *no face in F can be removed* **then**
11 Break
12 B=[0,...,$n_1$-1]
13 **for** $j = 0$ *to* $n_1 - 1$ **do**
14 $m_1$=flipnm B
15 **if** $e_1$ *has been removed* **then**
16 |A|-1 \\Initializes an array B[0... n1-1] of n1
17 **else** tetrahedrons, where e1 $\in$ e, and |A| is
18 return |A| reduced by 1 if e1 has been removed. If
 no edge in e can be removed, return |A|
19 **End**

**Fig 15. Tetrahedral optimization of the TetGen library to solve local optimum pseudo-code.**

by the creation of a 3D solid model through tetrahedral dissection of the closed spatial mesh model. Ultimately, a 3D finite element mesh is generated. Based on these steps, the following conclusions can be drawn.

(1) The intersecting fault zone consists of a fault surface grid, a stratigraphic interface grid, and a range grid. Analysis of the spatial relationships between these three types of grid surfaces leads to the conclusion that these grids exhibit intersecting and non-connected spatial relationships.

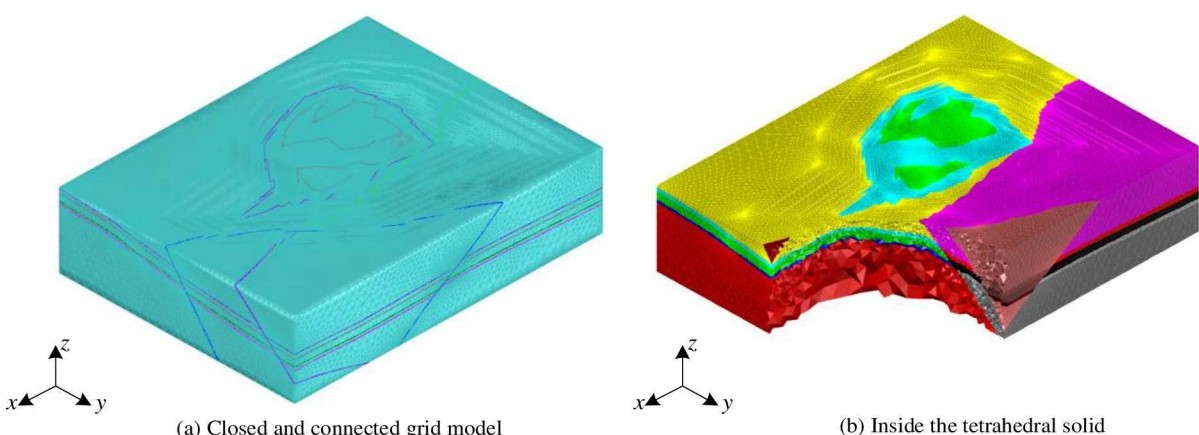

(a) Closed and connected grid model (b) Inside the tetrahedral solid

**Fig 16. Tetrahedral solid model generated by closed-connected mesh.**

**Table 1. Lithological information table.**

| No. | petrographic name | Cohesion c (*Mpa*) | Friction angle $\phi$ (°) | Capacity $\gamma$ (*KN/m³*) |
|-----|-------------------|--------------------|--------------------------|----------------------------|
| 1 | Topsoil | 0.0002 | 28.7 | 20.8 |
| 2 | Tertiary | 0 | 28.7 | 26.9 |
| 3 | Mudstone 1 | 0.05 | 25 | 19.6 |
| 4 | Coal | 0.1 | 30 | 12.2 |
| 5 | Mudstone 2 | 0.05 | 25 | 19.6 |

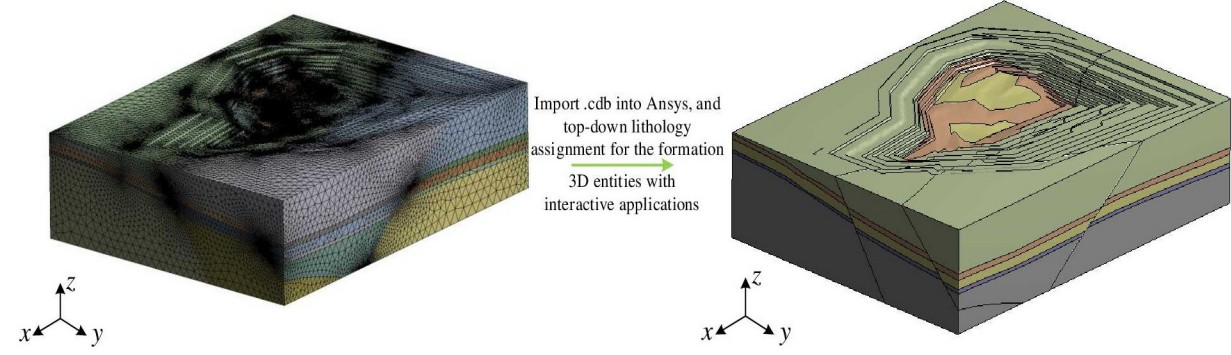

Import .cdb into Ansys, and top-down lithology assignment for the formation 3D entities with interactive applications

**Fig 17. Workflow for conversion to finite element mesh using Ansys.**

(2) According to the principle of spatial segmentation, a closed manifold method is proposed to achieve constraint processing on the intersecting grid surfaces of intersecting fault zones. A closed manifold spatial grid model comprising a geological fault surface grid, a ground surface grid, and a range grid is established, which offers a solution for handling the 3D complex stratigraphic relationships in 3D solid modeling.

(3) Based on the optimization of the closed mesh, an irregular tetrahedral solid model of the TetGen library-generated component region is used to successfully establish a 3D finite element mesh of the laminated geological body of the intersecting fault zone, which provides an effective and feasible solution for generating an accurate 3D finite element mesh for complex stratigraphic space.

(4) The three-dimensional geological model established in this paper not only simplifies the construction of the collection of intersecting faults but also realizes the spatial mechanical analysis of complex geology, which is easy to combine with the three-dimensional stability analysis to provide a scientific basis for the prevention and control of landslides and can be widely used in the complex three-dimensional geological modeling.

## Supporting information

**S1 Data.**
(RAR)

## Author Contributions

**Conceptualization:** YingXian Chen, HongXia Yang.

**Data curation:** YingXian Chen, JiaYing Li.

**Formal analysis:** YingXian Chen.

**Methodology:** HongXia Yang, YongChao Ye, JiaYing Li.

**Resources:** HongXia Yang.

**Software:** YingXian Chen.

**Supervision:** YingXian Chen.

**Validation:** HongXia Yang.

**Visualization:** HongXia Yang.

**Writing – original draft:** HongXia Yang, YongChao Ye, JiaYing Li.

**Writing – review & editing:** HongXia Yang.

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
