## [Decision Letter · Decision Letter 0]

9 Aug 2023

PONE-D-23-18613Generation of 3D finite element mesh of layered geological bodies in intersecting fault zonesPLOS ONE

Dear Dr. Yang,

Thank you for submitting your manuscript to PLOS ONE. After careful consideration, we feel that it has merit but does not fully meet PLOS ONE’s publication criteria as it currently stands. Therefore, we invite you to submit a revised version of the manuscript that addresses the points raised during the review process.

ACADEMIC EDITOR: Two reviewers have returned the reports where the writing issues has been noticed. The current manuscript contains publishable contribution however the writing needs to be further revised after a careful proof reading. The details of the proposed algorithm has not been presented clearly thus more details should be added in the revised version. A minor revision is reasonable to update the readability of this study.

We look forward to receiving your revised manuscript.

Kind regards,

Qichun Zhang, PhD

Academic Editor

PLOS ONE

Additional Editor Comments:

Two reviewers have returned the reports where the writing issues has been noticed. The current manuscript contains publishable contribution however the writing needs to be further revised after a careful proof reading. The details of the proposed algorithm has not been presented clearly thus more details should be added in the revised version. A minor revision is reasonable to update the readability of this study.

Reviewers' comments:

Reviewer's Responses to Questions

**Comments to the Author**

1. Is the manuscript technically sound, and do the data support the conclusions?

Reviewer #1: Yes

Reviewer #2: Yes

2. Has the statistical analysis been performed appropriately and rigorously? 

Reviewer #1: Yes

Reviewer #2: Yes

3. Have the authors made all data underlying the findings in their manuscript fully available?

Reviewer #1: Yes

Reviewer #2: Yes

4. Is the manuscript presented in an intelligible fashion and written in standard English?

Reviewer #1: Yes

Reviewer #2: Yes

5. Review Comments to the Author

Reviewer #1: The intersection relationship between faults and strata is of great significance for disaster prevention and oil exploration. In this paper, the authors propose a manifold processing method to establish the closed manifold spatial surface model of the intersecting fault zones, based on which the closed manifold spatial surface model of the intersecting fault zones and a three-dimensional solid model is established. I found this to be very interesting research and is important for us to understand the intersection relationship between faults and strata. The research will be of interest to the geological community. However, there are still some problems that need to be addressed. I propose to publish it after revision.Several sentences are not clear and need to be corrected.

Here are my comments.

1.Some odd white lines exist in Figure 8; please check and consider deleting them or explaining them.

2.The font in Figure 12 is consistent with the font in the other figures.

I propose to publish it after revision.

Reviewer #2: In this paper, a closed manifold processing method is proposed, which provides a solution for the generation of three-dimensional finite element mesh in complex formation space. The basic logic of the article is clear and the structure is reasonable. The article still has some aspects that can be improved. The reviewer suggested some changes to improve the article before it was published.

1. In the Introduction, the scientific and technological problems solved by 3D simulation of closed manifold are not discussed enough, the introduction of 3D finite element is not systematic enough, and the references are insufficient.

2.The paper uses a lot of space to describe the geological relationship between the intersecting fault zone and the closed manifold treatment, which is very clear and necessary. But the construction description of three-dimensional solid model is not detailed enough.

3.The Triangle 1.6 programs and TetGen Library are mentioned in the paper, and it is suggested that the relevant pseudocodes should be described in principle so that readers can have a clearer understanding of the article.

4.Conversion of the 3D solid model to finite element mesh in 3.2 is not introduced enough, too much process, it is recommended to add the principle introduction.

5.The conclusion should not simply repeat the research content, but should strengthen the scientific nature of the solution and the applicability of the technical method.

6. PLOS authors have the option to publish the peer review history of their article (what does this mean?). If published, this will include your full peer review and any attached files.

Reviewer #1: No

Reviewer #2: No

---

## [Author Response · Author response to Decision Letter 0]

6 Sep 2023

Dear Editors and Reviewers:

Thank you for your letter and the reviewers’ comments concerning our manuscript entitled “Generation of 3D finite element mesh of layered geological bodies in intersecting fault zones” (Manuscript Number: PONE-D-23-18613). Those comments are all valuable and very helpful for revising and improving our paper and the important guiding significance of our research. We have studied the comments carefully and tried our best to revise and improve the manuscript according to the constructive comments. We appreciate for Editors/Reviewers' warm work. We would be glad to respond to any further questions and comments that you may have. All the revised portions are marked in red in the article. The main corrections in the paper and the responses to the reviewers' comments are as follows:

Responses to the comments of Reviewer #1：

1.Comment: Some odd white lines exist in Figure 8.

Response: Thank you for your question. Explain the white color in Figure 8:The black line indicates the intersection line of the two faces, and the white dots indicate the required corresponding topology line of the face, we can judge whether the current topology of the two faces is consistent or not based on the white dots.

2.Comment: The font in Figure 12 is consistent with the font in the other figures

Response: Thank you very much for your correction of errors in this article. The font for Figure 12 has been modified to a consistent typeface.

The revised portion is marked in red in the Revised Manuscript with Track Changes.

Special thanks to you for your good comments

Responses to the comments of Reviewer #2：

1.Comment: In the Introduction, the scientific and technological problems solved by 3D simulation of the closed manifold are not discussed enough, the introduction of 3D finite element is not systematic enough, and the references are insufficient.

Response: Thank you for your question. As I understand, I have refined the Introduction section and relevant references have been added. The introduction and references have been updated with the following information:

For improved finite element analysis of geological bodies, precise mesh generation of the modeling area is necessary when constructing the geological solid model. In the modeling process, a complex geological space is comprised of numerous interconnected geological grid surfaces[28-30]. The correct topological connectivity relationship between each grid surface is typically a prerequisite for modeling, model editing, Boolean operations, or even 3D model analysis[31]. A. L. TERTOIS[32] et al employed discrete smoothing interpolation to develop a tool that enables small, real-time adjustments to faults in tetrahedral models. Qiang Tianchi[33] implemented a transition between 3D tetrahedral sparse and dense meshes by coupling two-part cells with an interface transition cell, using the principle of minimum potential energy to ensure coordination of the displacements and strains of each corresponding point on either side of the interface. Deyun Zhong[34] et al. introduced an adaptive mesh partitioning technique that separates the interstitial domain using feature constraints on the contour lines of the ore body or fits the intermediate domain using a distance field. Zhou[35]et al. studied the unstructured mesh generation method in finite element preprocessing and its application in mining, proposing the adaptive mesh segmentation method, which segregates or merges intermediate domains through the distance field and applies feature constraints to the contours of the ore bodies. Hang Si[36,37] et al. used the classical Delaunay refinement algorithm to build a TetGen library to generate an isotropic tetrahedral mesh. Souche L[38] et al. based on a global interpolation method for data extraction from faults and strata, capable of constraining any surface by all valid conformal data without being constrained by its type. Godefroy[39]used numerical fault operators to make the strata fall according to a theoretically isolated fault displacement model and produce consistent fault displacements, to make the structural model consistent. Lobatskaya[40] et al. used a finite element model to reconstruct the stresses within the block around the inclined fracture in the simulated fault damage zone. Feng[41] et al. determined the safety factor of the slope by creating a 3D numerical model and simulating the slope model with the finite element strength discount method, which allowed the identification of the specific damage mode of the slope during production. Sun Shiguo[42] et al. established a three-dimensional model calculated the displacement and stress of the slope body by the finite difference method, and investigated the characteristics of the influence of multi-stage open-pit cascade mining on the stability of the slope and its slope displacement and stress distribution. Chen[43] et al. created a three-dimensional solid model of the rock mass that was blasted by utilizing the lithology data as a sample. They then employed this model to compute the quantity of charge present in the blasted rock mass. Modeling individual faults was comparatively simple and straightforward. However, the modeling process becomes inherently complex when faults exhibit different tendencies, such as interlacing, overlapping, and truncation between fault surfaces. 

Additions to the references

[28]De Berg, M., Cheong, O., van Kreveld, M., & Overmars, M. (2008). Computational Geometry. doi:10.1007/978-3-540-77974-2 

[29]SCHNEIDER, P. J., AND EBERLY, D. H. 2002. Geometric Tools for Computer Graphics. Morgan Kaufmann Publishers.

[30]CHERPEAUN,CAUMONG,CAERSJ,etal.Method for stochastic inverse modeling of fault geometry and connectivity using flow data[J].Mathematical Geosciences,2012,44（2）:147-168.

[31]LIN Jianzhu,TANG Lei,YONG Junhai. Non-fluid closed triangle mesh regularisation for polygonal meshes[J]. Journal of Computer-Aided Design & Computer Graphics, vol.2014,26(10):1557-1566.

[32]TERTOISA,MALLETJ.Restoration of Complex Three-Dimensional Structural Models Based on the Mathematical Geo Chron Framework;proceedings of the81st EAGE Conferenceand Exhibition 2019,F,2019[C].European Association of Geoscientists&Engineers.

[33]Qiang Tianchi,Kou Xiaodong,Zhou Weiyuan. Three-dimensional finite element mesh encryption interface coordination method and its application in dam cracking analysis[J]. Chinese Journal of Rock Mechanics and Engineering, 2010,2000(05):562-566.

[34]ZHONG Deyun, WANG Liguan, BI Lin.Adaptive Meshing of Multi-domain Complex Orebody Models[J].Geomatics and Information Science of Wuhan University,2019,44（10）:1538-1544.

[35]Zhou Longquan. Research on unstructured tetrahedral mesh generation method and its application [D]. Qingdao: Shandong University of Science and Technology,2019.

[36]SI H. Adaptive tetrahedral mesh generation by constrained Delaunay refinement [J]. International Journal for Numerical Methods in Engineering, 2008, 75(7): 856-80.

[37]SI H. TetGen, towards a quality tetrahedral mesh generator [J]. 2013.https://doi.org/10.34657/3337

[38]SOUCHE L, ISKENOVA G, LEPAGE F, et al. Construction of structurally and stratigraphically consistent structural models using the volume-based modeling technology: Applications to an Australian dataset; proceedings of the International petroleum technology conference, F, 2014 [C]. International Petroleum Technology Conference.https://doi.org/10.2523/IPTC-18216-MS

[39]GODEFROY G, CAUMON G, FORD M, et al. A parametric fault displacement model to introduce kinematic control into modeling faults from sparse data [J]. Interpretation, 2018, 6(2): B1-B13.https://doi.org/10.1190/INT-2017-0059.1

[40]LOBATSKAYA R M, STRELCHENKO I P, DOLGIKH E S. Finite-element 3D modeling of stress patterns around a dipping fault [J]. Geoscience Frontiers, 2018, 9(5): 1555-63.https://doi.org/10.1016/j.gsf.2017.09.010

[41]FENG Cheng-gui,LIANG Wen-hai.Determination of Safety Factor of Slope Stability Based on Finite Element Numerical Simulation[J].Value Engineering,2023,42(12):95-98.

[42]SUN Shi-guo, SHAO Shu-sen, et al. lnfluence of Combined Method of Opencast and Underground Mining on Slope Stability[J]. Mining and Metallurgical Engineering,2020,40(5):15-18. DOI:10.3969/j.issn.0253-6099.2020.05.004.

[43]Chen Y X, Wang P F, Chen J, Zhou M,Yang H X,et al.Calculation of blast hole charge amount based on three-dimensional solid model of blasting rock mass[J].Scientific Reports,2022, 12(1):1-13.

[44]Liu Jia. Research on non-manifold surface transformation based on graphical rotation system [D]; Central South University, 2011.

[45]Li Zhaoliang, Pan Mao, Yang Yang, et al. 3D complex fault network modeling methods and applications [J]. Journal of Peking University (Natural Sciences Edition), 2015, 51(01): 79-85.DOI:10.13209/j.0479-8023.2014.173.

Additions highlighted in yellow

The revised portion is marked in red in the Revised Manuscript with Track Changes.

2.Comment: The construction description of the three-dimensional solid model is not detailed enough.

Response: Thank you for your question. After section 3.1, insert the following:

The generation of tetrahedral mesh in 3D solid modelling is the prerequisite and difficulty of FEA calculation, the specific modelling steps are as follows:(1) Establishment of a borehole database using engineering geological information in the region;(2) Grids of slope surfaces, geological interfaces, and fault surfaces were created through Delaunay triangular profiles interpolation using the inverse distance method;(3) Create a range grid and combine it with the slope surface grid to form a new range grid;(4) The new range grid is restricted to the level grid of each location, resulting in a closed, connected, and topologically consistent spatial surface domain;(5) Mesh optimization of the triangular mesh model underwent optimization using the Triangle 1.6 program;(6) Finally, using Visio Studio 2012 development platform, the TetGen library is developed to achieve the intersecting fault open pit mine slope mesh tetrahedral dissection to generate tetrahedral mesh, and the generated tetrahedral mesh is optimised.

Additions highlighted in yellow

The revised portion is marked in red in the Revised Manuscript with Track Changes.

3.Comment: The Triangle 1.6 programs and TetGen Library are mentioned in the paper, and it is suggested that the relevant pseudocodes should be described in principle so that readers can have a clearer understanding of the article.

Response: Thank you for your question. Triangle-optimized pseudo-code has been added following section 3.1.1. Pseudo-code for optimizing tetrahedrons has been added in section 3.1.2. The details are as follows:

Figure 11 Triangle optimization pseudo-code

The following has been added in 3.1.2:

A tetrahedral solid model generation and optimization program was developed using Visual C++2012 based on the TetGen library, and the tetrahedral optimization pseudo code is shown in Figure 13(a).To resolve the issue of local optima in the TetGen library tetrahedral optimization “local optima”, we integrate the procedure depicted in Figure 13(b) into the optimization process. The closed grid model of the intersecting fault zone was partitioned into tetrahedra to create an irregular tetrahedral solid model representing the intersecting fault zone, as shown in Figure 14 and Figure 15. Figure 16(a) illustrates the closed-connected grid model of the intersecting fault zone, while Figure 16(b) showcases the irregular tetrahedral solid model and its internal structure. Additionally, the tetrahedral solid models of different regions within the intersecting fault zone are distinguished using various colors.

Figure14 TetGen library for optimizing tetrahedral pseudo-code

Figure 15Tetrahedral optimization of the TetGen library to solve local optimum pseudo-code

Additions highlighted in yellow

The revised portion is marked in red in the Revised Manuscript with Track Changes.

4.Comment: it is recommended to add the principle introduction.

Response: Thank you for your question. Based on the issues raised, we have included the following information in section 3.2.

The three-dimensional geological solid model is combined through the interface of each rock layer to form a closed solid model, and the physical and mechanical parameters of the rock body are assigned to each geological entity, and the built irregular tetrahedral solid model is exported in the .cdb file format. The .cdb file was then imported into Ansys (Ansys Workbench Workbench, 2019) software for meshing, and the lithological parameter properties and contact properties of each layer were set to complete the construction of the 3D finite element model. Finally, the lithologies for each formation are assigned based on lithological parameters stated in Table 1. The lithology sequence, from top to bottom, includes topsoil, tertiary, mudstone, coal, and mudstone. This results in the conversion of the model into a 3D finite element mesh with interactive control, as shown in Figure 16.

Table 1 Lithological information table

No. petrographic name Cohesion c（Mpa） Friction angleφ（°） Capacity γ（KN/m3）

1 Topsoil 0.0002 28.7 20.8

2 Tertiary 0 26.9 20.8

3 Mudstone 1 0.05 25 19.6

4 Coal 0.1 30 12.2

5 Mudstone 2 0.05 25 19.6

Additions highlighted in yellow

The revised portion is marked in red in the manuscript.

5.Comment: The conclusion should not simply repeat the research content, but should strengthen the scientific nature of the solution and the applicability of the technical method.

Response: Thank you for your question. Based on the issues mentioned, we have revised the conclusion section as follows:

(2)According to the principle of spatial segmentation, a closed manifold method is proposed to achieve constraint processing on the intersecting grid surfaces of intersecting fault zones. A closed manifold spatial grid model comprising a geological fault surface grid, a ground surface grid, and a range grid is established, which offers a solution for handling the 3D complex stratigraphic relationships in 3D solid modeling.

(4) The three-dimensional geological model established in this paper not only simplifies the construction of the collection of intersecting faults but also realizes the spatial mechanical analysis of complex geology, which is easy to be combined with the three-dimensional stability analysis to provide a scientific basis for the prevention and control of landslides and can be widely used in the complex three-dimensional geological modeling.

Additions highlighted in yellow

The revised portion is marked in red in the Revised Manuscript with Track Changes.

Special thanks to you for your good comments

---

## [Decision Letter · Decision Letter 1]

8 Oct 2023

Generation of 3D finite element mesh of layered geological bodies in intersecting fault zones

PONE-D-23-18613R1

Dear Dr. Yang,

We’re pleased to inform you that your manuscript has been judged scientifically suitable for publication and will be formally accepted for publication once it meets all outstanding technical requirements.

Kind regards,

Qichun Zhang, PhD

Academic Editor

PLOS ONE

Additional Editor Comments:

The paper has been revised after a minor revision. One reviewer satisfied the revision and the other reviewer has not responded to the review invitation. As two of the reviewers only had minor concerns to the original submission while all the concerns have been addressed in the manuscript based on academic editor's personal reading, the current version can be accepted for publication.

Reviewers' comments:

Reviewer's Responses to Questions

**Comments to the Author**

1. If the authors have adequately addressed your comments raised in a previous round of review and you feel that this manuscript is now acceptable for publication, you may indicate that here to bypass the “Comments to the Author” section, enter your conflict of interest statement in the “Confidential to Editor” section, and submit your "Accept" recommendation.

Reviewer #2: All comments have been addressed

2. Is the manuscript technically sound, and do the data support the conclusions?

Reviewer #2: Yes

3. Has the statistical analysis been performed appropriately and rigorously? 

Reviewer #2: Yes

4. Have the authors made all data underlying the findings in their manuscript fully available?

Reviewer #2: Yes

5. Is the manuscript presented in an intelligible fashion and written in standard English?

Reviewer #2: Yes

6. Review Comments to the Author

Reviewer #2: In this paper, a closed manifold processing method is proposed, which provides a solution for the generation of three-dimensional finite element mesh in complex formation space. The basic logic of the article is clear and the structure is reasonable.

7. PLOS authors have the option to publish the peer review history of their article (what does this mean?). If published, this will include your full peer review and any attached files.

Reviewer #2: No

---

## [Editor Report · Acceptance letter]

14 Dec 2023

PONE-D-23-18613R1 

PLOS ONE

Dear Dr. Yang, 

I'm pleased to inform you that your manuscript has been deemed suitable for publication in PLOS ONE. Congratulations! Your manuscript is now being handed over to our production team.

Kind regards, 

on behalf of

Dr. Qichun Zhang 

Academic Editor

PLOS ONE